# A Study on the Effect of a Regular Exercise Habit on Health-Related Quality of Life in Adults with Cerebral Palsy

## Jungwan You

Health and Wellness College, Sungshin Women's University 2, 34 da-gil, Bomun-ro, Seongbuk-gu, Seoul 02844, Korea; rocggi@hanmail.net

**Abstract:** This study aimed to investigate the effects of exercise habits and types on the HRQOL of adults with CP, based on the fact that regular exercise habits, such as exercise, physical activity, etc., during leisure time have a positive effect on the improvement of HRQOL. A total of 215 adults (164 Men and 51 Women; 50 Gross Motor Function Classification System (GMFCS) Level I, 116 Level II, 24 Level III, and 25 Level IV) with CP but ambulant were recruited for this study. The mean (±SD) age of the subjects was 35.74 ± 9.65 years. They were evaluated using Medical Outcomes Study Short Form–36 (SF-36), the world's most popular questionnaire for measuring health status, which is used widely in research targeting all healthy and elderly individuals with disabilities and is also used for the evaluation of HRQOL. SF-36 includes 8 subscales with 36 questions. In addition, in order to investigate further into exercise habits and types, four more detailed questions were asked: presence of exercise habits at least once a week; duration of exercise in adults who have exercise habits with CP; weekly exercise frequency; time of a single exercise session. The following are the results of the analysis of variance to confirm the difference in HRQOL depending on the presence of exercise habits in adults with CP. A significant difference was found in all items, except for the BP item, among the subscales of the HRQOL ($p < 0.01$), with higher values observed for those with exercise habits compared with those with no exercise habits. A significant difference was found in all items, except for BP, among the subscales of HRQOL ($p < 0.01$). Additionally, a positive (+) correlation was found in RP, VT, RE, MH, and MCS ($p < 0.05$) in terms of exercise duration in adults who have exercise habits with CP. In addition, a negative (−) correlation was found in SF, RE, and MCS ($p < 0.05$) in terms of the weekly exercise frequency. However, no significant correlation was found in all subscales of HRQOL in terms of the time of a single exercise session. In conclusion, it has become clear from adults with CP that engaging in regular exercise at least once a week has a positive effect on improving all subscales of HRQOL except for BP. Efforts by individuals, groups, and families all need to be made in order for adults with CP to have regular exercise habits to improve their HRQOL.

**Keywords:** cerebral palsy; health-related quality of life; exercise habit; duration of exercise; weekly exercise frequency



## 1. Background

CP (Cerebral Palsy) can be defined as a permanent disability group that restricts the movement and posture of people, caused by non-progressive disturbances that may occur in the processes of brain growth in a fetus or an infant [1]. The World Health Organization has announced that movement disorders in CP are accompanied by sensory disorders, perceptual disorders, cognitive disorders, communication disorders, and behavior disorders mainly due to musculoskeletal problems [2,3]. Therefore, it can be stated that CP is not a single disease, but a heterogeneous and complex one characterized by abnormal movement patterns and postures [4], because a variety of disorders, including movement disorders, etc., occur due to non-progressive brain lesions [5]. Among them, movement disorders



directly related to the deterioration of physical functions can be considered as the most common clinical signs of CP.

Those with physical disabilities, such as CP, have limited social activities compared with those who do not, so not only do they experience more pain, sadness, anxiety, and insomnia [6], but they also have a higher risk of cardiovascular disease, diabetes, and cancer [7]. In addition, the limited social activities resulting from physical problems adversely affect health, thereby lowering their quality of life (QOL) [8]. QOL refers to subjective satisfaction, well-being, and happiness regarding the cognitive and emotional levels of individuals who feel the objective elements of life based on emotional well-being, personal relations, material well-being, personal development, physical well-being, self-determination, and social integration and rights [9]. QOL is becoming an emerging concept, which is not only used for ordinary people, but also for many parts of the services in the welfare field for the disabled, and is used as data to predict social and emotional adaptive behaviors [10–12] and to evaluate the therapeutic effect on health and mental status [13,14].

In general, the limited physical movements observed in adults with CP and the resulting restriction of movement are thought to cause problems in their daily lives and social lives that adversely affect their QOL. Personally, adults with CP are required to receive medical assistance and protection, and to be taken care of by their families from childhood, increasing stress on the families and further negatively affecting their mental state [15]. The Gross Motor Function Classification System (GMFCS) that represents the degree of motor disorders in adults with CP is used to investigate its effect on their lives [1,13,15–17]. Study results have reported that the GMFCS, which can represent the exercise ability of adults with CP, has a close relationship with QOL and HRQOL, specifically because the higher the level of GMFCS, the lower the QOL and HRQOL, and that in particular, they are strongly associated with physical happiness, rather than psychological happiness [16,17]. In addition, a number of limitations, including discomfort, etc., in the daily and social lives of adults with CP adversely affect their health status, which may also lead to a suggestion for community participation or to a lowered QOL [18]. Eventually, they might grow negative ideas that could cause lowered QOL, blaming their disability, which would put them at risk of becoming increasingly isolated from society [19,20]. In other words, it is considered desirable to evaluate QOL for predicting both the health status and disease of those who have disabilities, such as CP, etc., and their social and emotional adaptive behaviors.

In the context of increasing interest in the functional and mental aspects of those with disabilities and the types of and changes in disabilities in recent years, one of the areas of QOL to focus only on those aspects is the health-related quality of life (HRQOL). The HRQOL evaluates the effect before and after treatment on the state of their well-being [18], and can predict any disease they may have or their current health status [12].

There are many factors that affect QOL, such as health status, lifestyle, living environment, and job status. Additionally, HRQOL can be largely divided into two types, i.e., the physical and mental aspects, and it is thought that the most important factor in improving HRQOL is moderate or higher physical activity. Among such physical activities, exercise is considered to be the most important among health behaviors, and it has been reported that regular exercise habits have a significant effect on the improvement of HRQOL according to the results of previous studies [21–23]. There have been no studies that identified the level of physical or sports activity in adults with CP, but for children, adolescents with typical development, and healthy adults, recommendations call for a training frequency of two to three times per week on nonconsecutive days [24]. Studies on the relationship between the exercise habits of adults with CP and their HRQOL have not yet been conducted. However, it has been reported that the depression-related score of elderly people with depression becomes significantly lower at the end of exercise than that before exercise, according to the results of previous studies on chronically ill patients [19], stress and depression in adults with cardiovascular risk factors are reduced, and their HRQOL is improved according to a study on the effect of Tai Chi and Qigong medical exercises on those adults. It has also

been reported that exercise and physical activity during leisure time have a positive effect on the improvement of HRQOL according to a study on general adults [23]. As such, it can be thought that a regular exercise habit is closely related to improving HRQOL, and that the habit is also an important factor in improving HRQOL of adults with CP. However, studies on the relationship between the exercise habits of adults with CP and their CP have not yet been conducted.

Therefore, this study aimed to investigate the effect of exercise habits and types on the HRQOL of adults with CP based on the fact that regular exercise habits, such as exercise, physical activity, etc., during leisure time have a positive effect on the improvement of the HRQOL.

## 2. Methods

This study conforms to the Declaration of Helsinki. All subjects gave written, informed consent before participating in the study. As this study was not an experimental study on people, but was simply a study based on a questionnaire, it did not need to be approved by the Institutional Review Board of Sungshin Women's University.

### 2.1. Participants

This study was conducted from April 2015 to December 2019 at the Seoul Municipal CP Welfare Center in Korea and at the Hiroshima and Hyogo Rehabilitation Center in Japan, and a total of 215 (164 Men and 51 Women) adults with CP but ambulant were recruited for this study. A total of 118 adults with CP had exercise habits (50 for Soccer, 20 for Bossia, 18 for Track and Field, 18 for Sitting Volley-ball, and lastly 12 for Swimming), and 40 had GMFCS Level I, 50 had Level II, 12 had Level III, and 11 Level IV. However, 97 adults had CP and were without exercise habits (10 with GMFCS Level I, 61 with Level II, 12 with Level III, and lastly 14 with Level IV). The mean ($\pm$SD) age of the subjects was $35.74 \pm 9.65$ years.

### 2.2. HRQOL Measure

The Korean and Japanese versions of the Medical Outcomes Study Short Form–36 (SF-36), the world's most popular questionnaire for measuring health status, are used widely in research targeting all healthy individuals and elderly individuals with disabilities and are also used for the evaluation of the HRQOL.

SF-36 includes the following 8 subscales with 36 questions: physical functioning (PF), role physical (RP), bodily pain (BP), general health (GH), physical component summary (PCS) and social functioning (SF), vitality (VT), role emotional (RE), mental health (MH), and mental component summary (MCS). Additionally, assessing the eight subscale scores, PCS and MCS enable a more comprehensive evaluation of the HRQOL. The higher the subscale's score, the more positively it affects the HRQOL. All surveys were conducted through one-on-one interviews between the researcher and each of the study subjects.

### 2.3. Exercise Habits and Types

In order to further investigate exercise habits and types, four more detailed questions were asked: presence of exercise habits at least once a week (① Yes or ② No); duration of exercise in adults who have exercise habits with CP (① 1 year or more, ② more than 5 years, ③ more than 10 years, or ④ more than 15 years); weekly exercise frequency (① once or more, ② more than 2 times, ③ more than 3 times, or ④ more than 4 times); and time of a single exercise session (① 1 h or more, ② more than 2 h, ③ more than 3 h, or ④ more than 4 h).

*2.4. Data Analysis*

Descriptive statistics, including the mean and SDs, were calculated for all variables. Statistical analyses were performed using SPSS for Windows V20.0. A one-way analysis of variance was used to compare the HRQOL scores of the participants in terms of exercise habits and their degree of exercise habits. The relationship between the degree of exercise habits and HRQOL was analyzed by Pearson's correlation coefficient. The significance level was set at 5%.

## 3. Results

The following are the results of the analysis of variance to confirm the difference in the HRQOL depending on the presence of exercise habits in adults with CP (Table 1). A significant difference was found for all items, except the BP item, among the subscales of the HRQOL ($p < 0.01$), with higher values observed for those with exercise habits compared with those with no exercise habits.

Analysis of variance was conducted to determine what difference in HRQOL occurs according to the types of exercise habits in adults with CP (Tables 2–4). Additionally, a significant difference was found only in MH among all subscales of HRQOL in terms of exercise duration ($p < 0.05$) (Table 2). In terms of the weekly exercise frequency, a significant difference was found only in SF among all subscales of the HRQOL ($p < 0.05$) (Table 3). The higher weekly exercise frequency had a more positive effect on the SF of HRQOL. Lastly, in the case of time for the single exercise session, it was found not to affect all subscales of HRQOL (Table 4).

**Table 1.** Degree of HRQOL according to the presence of exercise habits.

| | PF | RP | BP | GH | VT | SF | RE | MH | PCS | MCS |
|---|---|---|---|---|---|---|---|---|---|---|
| Exercise habits (n = 118) | 78.21 ± 22.74 | 76.07 ± 22.21 | 70.10 ± 24.89 | 64.22 ± 17.69 | 62.52 ± 20.28 | 79.02 ± 21.79 | 80.29 ± 22.58 | 68.68 ± 18.71 | 72.23 ± 14.65 | 72.63 ± 16.27 |
| No exercise habits (n = 97) | 52.12 ± 31.14 | 56.65 ± 25.63 | 65.65 ± 28.85 | 50.81 ± 19.12 | 46.92 ± 21.71 | 67.78 ± 24.98 | 57.90 ± 27.70 | 56.39 ± 21.32 | 56.31 ± 15.73 | 57.25 ± 18.67 |
| F | 50.157 | 35.383 | 1.676 | 28.438 | 29.527 | 12.404 | 42.643 | 20.259 | 58.717 | 41.597 |
| *p* | **0.000 ***** | **0.000 ***** | 0.197 | **0.000 ***** | **0.000 ***** | **0.001 **** | **0.000 ***** | **0.000 ***** | **0.000 ***** | **0.000 ***** |

** $p < 0.01$, *** $p < 0.001$.

**Table 2.** Degree of HRQOL depending on exercise duration.

| | PF | RP | BP | GH | VT | SF | RE | MH | PCS | MCS |
|---|---|---|---|---|---|---|---|---|---|---|
| 1 year or more (n = 36) | 80.24 ± 21.65 | 68.76 ± 23.52 | 70.08 ± 23.82 | 62.52 ± 19.35 | 58.87 ± 20.66 | 76.38 ± 21.08 | 73.38 ± 24.21 | 66.25 ± 20.74 | 70.40 ± 15.57 | 68.72 ± 17.36 |
| More than 5 years (n = 38) | 79.07 ± 22.74 | 78.13 ± 23.45 | 73.52 ± 25.41 | 63.10 ± 13.33 | 59.56 ± 19.54 | 81.25 ± 21.10 | 81.13 ± 22.72 | 63.42 ± 18.67 | 73.46 ± 12.77 | 71.34 ± 15.41 |
| More than 10 years (n = 16) | 69.68 ± 26.48 | 80.87 ± 20.07 | 74.43 ± 22.59 | 66.68 ± 21.94 | 67.60 ± 24.17 | 78.90 ± 23.14 | 85.93 ± 18.19 | 71.87 ± 16.11 | 72.92 ± 17.22 | 76.07 ± 17.25 |
| More than 15 years (n = 28) | 79.28 ± 22.14 | 79.92 ± 18.34 | 64.28 ± 26.80 | 66.53 ± 18.57 | 68.32 ± 17.33 | 79.46 ± 23.62 | 84.82 ± 21.27 | 77.14 ± 14.49 | 72.50 ± 14.86 | 77.44 ± 14.56 |
| F | 0.879 | 1.987 | 0.903 | 0.417 | 1.798 | 0.306 | 1.893 | 3.474 | 0.285 | 1.866 |
| *p* | 0.454 | 0.120 | 0.442 | 0.741 | 0.153 | 0.851 | 0.135 | **0.018 *** | 0.836 | 0.139 |

* $p < 0.05$.

**Table 3.** Degree of HRQOL depending on the weekly exercise frequency.

| | PF | RP | BP | GH | VT | SF | RE | MH | PCS | MCS |
|---|---|---|---|---|---|---|---|---|---|---|
| Once or more (n = 69) | 75.70 ± 25.71 | 79.81 ± 20.05 | 72.39 ± 24.75 | 66.15 ± 16.51 | 62.61 ± 18.79 | 83.33 ± 18.77 | 84.17 ± 19.44 | 71.15 ± 16.60 | 73.51 ± 14.58 | 75.32 ± 13.92 |
| More than 2 times (n = 25) | 81.40 ± 15.37 | 70.01 ± 22.52 | 66.20 ± 25.53 | 64.20 ± 17.00 | 66.77 ± 19.82 | 78.50 ± 22.39 | 77.36 ± 21.18 | 65.00 ± 17.79 | 70.45 ± 13.94 | 71.30 ± 15.55 |
| More than 3 times (n = 15) | 85.00 ± 15.11 | 72.92 ± 30.30 | 62.80 ± 26.83 | 55.00 ± 24.21 | 61.27 ± 25.78 | 65.00 ± 25.53 | 69.44 ± 30.80 | 66.00 ± 27.59 | 68.93 ± 17.75 | 65.43 ± 21.83 |
| More than 4 times (n = 9) | 77.22 ± 25.26 | 69.45 ± 19.11 | 79.55 ± 18.68 | 64.88 ± 13.19 | 52.08 ± 22.09 | 70.83 ± 26.51 | 76.85 ± 29.08 | 64.44 ± 19.11 | 72.75 ± 12.40 | 66.05 ± 21.34 |
| F | 0.889 | 1.668 | 1.265 | 1.667 | 1.186 | 3.626 | 2.105 | 0.982 | 0.551 | 2.177 |
| *p* | 0.449 | 0.178 | 0.290 | 0.178 | 0.318 | **0.015 *** | 0.104 | 0.404 | 0.648 | 0.095 |

* $p < 0.05$.

**Table 4.** Degree of HRQOL according to the time of a single exercise session.

| | PF | RP | BP | GH | VT | SF | RE | MH | PCS | MCS |
|---|---|---|---|---|---|---|---|---|---|---|
| 1 h or more (n = 7) | 78.57 ± 15.99 | 60.72 ± 11.83 | 64.57 ± 20.80 | 63.57 ± 14.17 | 47.34 ± 11.89 | 83.92 ± 15.66 | 71.41 ± 20.30 | 64.28 ± 10.57 | 66.86 ± 8.20 | 66.74 ± 10.18 |
| More than 2 h (n = 45) | 72.55 ± 23.87 | 78.48 ± 23.24 | 70.13 ± 26.27 | 63.91 ± 18.71 | 64.04 ± 24.64 | 76.94 ± 22.91 | 80.92 ± 23.54 | 69.11 ± 21.14 | 71.27 ± 15.28 | 72.75 ± 19.20 |
| More than 3 h (n = 50) | 82.37 ± 20.88 | 74.64 ± 20.94 | 71.60 ± 24.58 | 64.94 ± 17.34 | 62.52 ± 16.36 | 79.50 ± 21.24 | 79.66 ± 20.35 | 68.50 ± 18.49 | 73.39 ± 14.71 | 72.54 ± 14.34 |
| More than 4 h (n = 16) | 80.93 ± 26.15 | 80.48 ± 24.87 | 70.00 ± 25.31 | 63.18 ± 18.64 | 64.86 ± 19.35 | 81.25 ± 23.71 | 84.38 ± 27.69 | 70.00 ± 15.70 | 73.65 ± 15.25 | 75.12 ± 15.71 |
| F | 1.583 | 1.594 | 0.166 | 0.052 | 1.481 | 0.313 | 0.554 | 0.161 | 0.526 | 0.425 |
| *p* | 0.197 | 0.195 | 0.919 | 0.984 | 0.223 | 0.816 | 0.647 | 0.922 | 0.666 | 0.735 |

## 4. Discussion

This study investigated the effect of regular exercise habits on the HRQOL in adults with CP and analyzed the relationship between the exercise types of the adults who have exercise habits with CP and their HRQOL.

The analysis results showed that adults with CP who had regular exercise habits of at least once a week had significant differences in all items, except BP, compared with those who did not ($p < 0.01$), suggesting that regular exercise habits have a positive effect on the HRQOL of adults with CP.

The results of this study were in keeping with results from previous studies reporting that exercise and physical activities have a positive effect on the HRQOL of adults with CP and other disabled people [8,25,26]. The results of a survey on the effect of recreational exercise on the QOL of 129 disabled people participating in adapted sports programs concluded that participation in the programs had a positive effect on individuals' perceptions of health and well-being, reporting that participation in the programs alone causes positive changes in health (79%), QOL (70%), and family life (69.4%) [25]. In addition, other studies have suggested a mechanism by which individuals may feel happiness or rewarded by participating in adapted sports programs, and have reported that they may feel a sense of accomplishment or a sense that their body is in motion through regular physical activity, having a positive effect on mental well-being that increases self-efficacy, self-confidence, and self-esteem [27]. Therefore, such studies have made it clear that, for adults with CP who are able to express their emotions through sports, regular participation in sports has a positive effect on their HRQOL.

This study has shown that the presence of exercise habits does not have a significant effect on BP among all subscales of HRQOL. Most adults with CP live their entire lives with chronic pain, so it is one of the most common personal complaints [16,28,29]. One study reported that adults with CP who experience chronic pain are often forced to change their way of life, which can eventually be considered a defeat and then cause psychological distress to them [17]. This study has shown that the presence of exercise habits has the least effect on BP in adults with CP, supporting results from previous studies reporting no significant relationship between chronic pain and QOL [8]. In the end, it is considered that a regular exercise habit is not a factor that can solve chronic pain, such as BP, in the HRQOL.

The results of correlation analysis for determining the type of relationship between the types of exercise habits of adults with CP and their HRQOL have shown positive (+) correlations with RP only among the physical aspects and with VT, RE, MH, and MCS, excluding SF, among the mental aspects, regarding all subscales of the HRQOL, in terms of exercise duration ($p < 0.05$), while the results of the analysis of variance showed a significant difference for MH ($p < 0.05$). Results from previous studies reported that the higher the level of GMFCS, the lower the QOL and HRQOL, and that there was a stronger correlation among the physical aspects, rather than mental ones [30,31]. Analyzing the two studies, it would be a natural result that adults with CP who suffer a severe degree of disability feel dissatisfaction in terms of physical aspects. However, it is important to note that, even if the degree of disability was not very serious, there was no increase in satisfaction in mental aspects. On the contrary, the results of this study showed that exercise duration had a closer relationship with mental aspects, rather than physical ones, in adults with CP. In other words, with regard to the HRQOL of adults with CP, it has become clear that maintaining regular exercise habits, rather than GMFCS indicating the level of motor functions, is an important factor for improving mental aspects, and that as the duration of exercise increases, the more positively it affects MH.

In the case of the weekly exercise frequency, it exhibited negative (−) correlations only in SF, RE, and MCS among the mental aspects regarding all subscales of HRQOL ($p < 0.05$), while the results of the analysis of variance showed a significant difference with SF ($p < 0.05$). The results of previous studies reported that adults with CP lose their physical strength more rapidly than the general public [32–34] due to the natural aging process (i.e., decrease in strength and stamina), as well as changes regarding their condition (i.e.,

decrease in mobility and cramps) [35,36]. In addition, adults with CP suffer from premature senility and other complications [37], which makes it difficult for them to work and go out in a social environment [37,38]. To make matters worse, it has a huge impact on interfering with their social life and lowering their activities of daily living (ADL) and QOL [1,3,39–41]. These studies lead us to believe that the higher the exercise frequency, the more positive the effects it will have on the mental aspects of HRQOL.

No significant correlation was found in all subscales of HRQOL in terms of the time of a single exercise session. However, numerically, it was found that the time of a single exercise session had a positive effect on the PCS and MCS of HRQOL when it was about 2 h.

## 5. Conclusions

In this study, it became clear from adults with CP that regular exercise at least once a week has a positive effect on improving all subscales of HRQOL, except for BP. The study results showed that the longer the duration of exercise, the more closely related it was to improving the mental aspects of the HRQOL, rather than the physical aspects, in adults with CP who had exercise habits, which was thought to have a positive effect on the improvement of MH in particular. In addition, the study results showed that, numerically, the weekly exercise frequency was more closely related to the improvement in the mental aspects of HRQOL when it was 1 to 2 times a week than when it was 3 to 4 times a week or more, which was thought to have a positive effect on the improvement in SF in particular. In conclusion, efforts by individuals, groups, and families all need to be made in order for adults with CP to have regular exercise habits to improve their HRQOL.

**Funding:** This research received no external funding.

**Institutional Review Board Statement:** This study conforms to the Declaration of Helsinki. As this study was not an experimental study on people, but was simply based on a questionnaire, it did not need to be approved by the Institutional Review Board. The investigation conforms to the Declaration of Helsinki.

**Informed Consent Statement:** All subjects gave written, informed consent before participating in the study.

**Data Availability Statement:** Not available.

**Conflicts of Interest:** The authors declare no conflict of interest.

## Abbreviations

CP: Cerebral palsy; GMFCS: Gross Motor Function Classification System; QOL: Quality of Life; HRQOL: Health-Related Quality of Life; PF: physical functioning; RP: role physical; BP: bodily pain; GH: general health; PCS: Physical component summary; SF: social functioning; VT: vitality; RE: role emotional; MH: mental health; MCS: mental component summary.

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
