# Peer review of "A Study on the Effect of a Regular Exercise Habit on Health-Related Quality of Life in Adults with Cerebral Palsy"

_applsci, doi:10.3390/app12189068_

Round 1

Reviewer 1 Report

The large number of respondents with cerebral palsy provides valuable data on the relationship between their exercise habit and QOL. However, this paper needs to be improved such as data presentation before publication.

<Major comments>

I recommend to delete Table 2 because three independent variables in Table 2 are analyzed in other Tables (Table 3-5). In addition, their variables are sequential scales so that Pearson’s correlation analysis is inappropriate.

This study focuses on Exercise habit in adults with CP, but there is no explanation of what kind of exercise participants are doing.

<Minor comments>

Abstract and Results sections:

This paper is frequently using “there was” in the Result section. Please replace other expressions

Methods:

2) HRQOL Measure

Please describe whether a higher score of subscale is better or worse for participant’s health.

Results:

In all tables, significant digits are too many.

Author Response

Thank you for insight on the important part of my research.

We have correctly revised the manuscript where you have pointed out. 

Thank you once again.

We had an American native English proofread the paper once more.

We greatly appreciate your advice and we will continue to make great efforts for CP research under insufficient circumstances

Reviewer 2 Report

The study aimed to investigate the effect of exercise habits and types on health related quality of life (HRQOL) of adults with CP. A total of 215 adults with cerebral palsy (CP) were evaluated by the Medical Outcomes Study Short Form (SF-36). The results showed that regular exercise enrollment at least once a week has a positive effect on improving almost all the subscales of HRQOL). Moreover, the longer the duration of exercise, improvement of mental aspects was also observed.

In general, this study aims on an important thematic and the one that is of largely interest. In addition, the manuscript is relatively well organized and structured. Despite of this, there are several issues that should be clarified and further defined in the present form that the manuscript was presented.

The first issue is that the manuscript needs to be revised regarding language usage. There are many cases that the manuscript is hard to be followed or that the sentence is awkward. I will point a few cases (minor aspects) but still a revision would be very welcome.

The second issue is related to the conclusion of the study, mainly, in the abstract lines 27-30) and at the end of the conclusion (252-256). The message stated in both of these cases does not reflect and are not based upon the results of the study.

 The third issue is related to the presentation and interpretation of the results. I will point specifically each of the cases for the RESULTS:

Line 140: it would be interesting to be more direct in the presentation. For instance, I would suggest the sentence a follow: “There was a significant difference in all items, except the BP item, among the subscales of the HRQOL (p<0.01), with higher values observed for those with exercise habits compared to those with no exercise habits.”

Line 160: SF was also significantly different among frequency of exercise. Please describe the difference.

Line 143: please further elaborate the information in the legend. Could mention the subscales and the exercise habits, avoiding that the reader needs to go back in the manuscript to see what each subscales are. The same would be applied to all other legends. Please include all the necessary information in order to the reader look at the table and understand without going to the text.

Line 145: Table 2 does not show results of analysis of variance, but Person’s correlation coefficients

Lines 160: what is meant by “numerically”? Was the mentioned effect of weekly exercise statistically significant for PCS and MCS? If not, there is nothing to say about it. If there is no significant difference, discussion of this issue (lines 228-238) is not pertinent. If there is significant difference, such difference statistical results should be presented.

Minor aspects:

Line 10: include “adults” after 215

Lines 27-30: conclusion should be re-phrase because these are not the results of the study

Lines 74-77: what is the conveyed information in this phrase? What does “religious beliefs” mean here? Is this phase necessary?

Lines 101-104: Please clarify if this is a local procedure regarding the usage of questionnaire. In many places, even studies only using questionnaire need to be approved by the Institutional Review Board

Line 171: I guess here would require a subtitle “4.Discussion”

Line 177: please revise the use of contraction form (didn’t) by “did not”. Please apply such use throughout the manuscript

Line 199: please include “results from” in the phrase “… supporting results from previous studies …” Please apply this form throughout the manuscript.

Line 228-238: please revise the information here according to the previous comment

Line 252-256: revise the conclusion as mentioned previously.

Author Response

Thank you for insight on the important part of my research.

We have correctly revised the manuscript where you have pointed out. 

Thank you once again.

We greatly appreciate your advice and we will continue to make great efforts for CP research under insufficient circumstances.

Reviewer 3 Report

Abstract
The description of the information obtained seems unintelligible: 'Presence of exercise habits at least once a week; Duration of exercise in adults who have exercise habits with CP; Weekly exercise
frequency; Time for one exercise session."
The abbreviations in the description of the results make the abstract difficult to understand.

Keywords: all occur in the title of the text.

Background: the general theoretical basis for cerebral palsy is completely redundant. It would be necessary to focus on the specifics of motor activity in adults with CP.

Methods:
(a) throughout the text, there is mention of exercise habits and types, but only time and frequency were surveyed - there is no mention of any breakdown by type or level of cardiovascular load.
b) the time options are illogical - 1) once or more, more than twice - could this have been misleading for probands?
(c) movement habits were collected by verbal questioning? Who conducted it? who assessed the SF-36?
d) I would have liked a more detailed characterisation of the cohort with respect to the type of cerebral palsy.

Table 1: what does "No Absence of exercise habits" mean?
Table 2: why is "monthly exercise frequency" mentioned when weekly frequency was surveyed?

The results flow freely into the discussion, which the text does not use as a section.

I would welcome a table showing the distribution of frequency and duration of exercise relative to GMFCS level.
What was taken as "exercise"? Was the intensity of exercise considered?

Author Response

Thank you for insight on the important part of my research.

We have correctly revised the manuscript where you have pointed out. 

Thank you once again.

Further I would like to request for your understanding in respect to the review points 

(a) throughout the text, there is mention of exercise habits and types, but only time and frequency were surveyed - there is no mention of any breakdown by type or level of cardiovascular load.

-This researcher also wanted to investigate the health status of adults with CP, including the level of the cardiovascular system. However, there was great difficulty with a one-on-one interview with adults with CP who had a intellectual level significantly lower than that of normal people to clearly investigate their diseases. I will try to investigate the level of the cardiovascular system in a follow-up study.

(b) the time options are illogical - 1) once or more, more than twice - could this have been misleading for probands?

-No form was developed to investigate the exercise habits of adults with CP, so that I referred to the questionnaire used in the similar previous studies. Next time, I will give careful consideration to that point.

I would welcome a table showing the distribution of frequency and duration of exercise relative to GMFCS level.

-It has already been verified by previous studies that the higher the GMFCS level of adults with CP, the freer their movements, having a positive effect on their domains of physical activity, including the frequency and duration of exercise, etc. And, I added it to the introduction of this study.

What was taken as "exercise"? Was the intensity of exercise considered?

-Exercise intensity is very important to the general public, but for adults with CP who have difficulty in moving their body, participation in and execution of exercise itself is considered to be the amount of high-intensity physical activity, and it is considered to be very difficult in reality to investigate adults with CP for subjective judgment on exercise intensity through oral questions, as mentioned above, so the exercise intensity was excluded in this study.

We greatly appreciate your advice and we will continue to make great efforts for CP research under insufficient circumstances.

Round 2

Reviewer 1 Report

Appropriate revisions have been made based on the comments.